

# An adapted deep convolutional RNN model for spatio-temporal prediction of wind speed extremes in the short-to-medium range for wind energy applications

Daan Scheepens[1], Irene Schicker[2], Kateřina Hlaváčková-Schindler[1], and Claudia Plant[1]

[1]Research Group Data Mining and Machine Learning, Faculty of Computer Science, University of Vienna, Währingerstrasse 29, 1090 Vienna, Austria; d.r.scheepens@gmail.com (DS); katerina.schindlerova@univie.ac.at (KHS); claudia.plant@univie.ac.at (CP).
[2]Zentralanstalt für Meteorologie und Geodynamik (ZAMG), Hohe Warte 38, 1190 Vienna, Austria; irene.schicker@zamg.ac.at (IS)

**Correspondence:** d.r.scheepens@gmail.com, irene.schicker@zamg.ac.at

**Abstract.** The amount of wind farms and wind power production in Europe, both on- and off-shore, has increased rapidly in the past years. To ensure grid stability, on-time (re)scheduling of maintenance tasks and mitigate fees in energy trading, accurate predictions of wind speed and wind power are needed. It has become particularly important to improve wind speed predictions in the short range of one to six hours as wind speed variability in this range has been found to pose the largest operational

challenges. Furthermore, accurate predictions of extreme wind events are of high importance to wind farm operators as timely knowledge of these can both prevent damages and offer economic preparedness. In this work we propose a deep convolutional recurrent neural network (RNN) based regression model, for the spatio-temporal prediction of extreme wind speed events over Europe in the short-to-medium range (12 hour lead-time in one hour intervals). This is achieved by training a multi-layered convolutional long short-term memory (ConvLSTM) network with so-called imbalanced regression loss. To this end

we investigate three different loss functions: the inversely weighted mean absolute error (W-MAE) loss, the inversely weighted mean squared error (W-MSE) loss and the squared error-relevance area (SERA) loss. We investigate forecast performance for various high-threshold extreme events and for various numbers of network layers, and compare the imbalanced regression loss functions to the commonly used mean squared error (MSE) and mean absolute error (MAE) loss. The results indicate superior performance of an ensemble of networks trained with either W-MAE, W-MSE or SERA loss, showing substantial

improvements on high intensity extreme events. We conclude that the ConvLSTM trained with imbalanced regression loss provides an effective way to adapt deep learning to the task of imbalanced spatio-temporal regression and its application to the forecasting of extreme wind events in the short-to-medium range, and may be best utilised as an ensemble. This work was performed as a part of the MEDEA project, which is funded by the Austrian Climate Research Program to further research on renewable energy and meteorologically induced extreme events.



## 1  Introduction

Global warming demands ever more urgently that electricity generation is shifted away from fossil fuels and towards renewable energy sources. Although global demands for fossil fuels are not yet showing signs of decreasing, renewables are on the rise. In 2021, more than half of the growth in global electricity supply was provided by renewables, while the share of renewables in global electricity generation reached close to 30 %, having steadily risen over the past decades (IEA, 2021). Possessing

the largest market share among the renewables, wind energy has managed to establish itself as a mature, reliable and efficient technology for electricity production and is expected to maintain rapid growth in the coming years (Fyrippis et al., 2010; Huang et al., 2015). Thanks to continued advancements in on- and offshore wind energy technology and the associated continued reduction in costs, wind power capacity could grow from having met 1.8 % of global electricity demand in 2009 to meeting roughly 20 % of demand in 2030 (Darwish and Al-Dabbagh, 2020). Indeed, many countries have already demonstrated that

hybrid electric systems with large contributions of wind energy can operate reliably. For example, in as early as 2010, Denmark, Portugal, Spain and Ireland managed to supply between 10 and 20 % of annual electricity demand with wind energy (Wiser et al., 2011) and the numbers have only risen since.

One of the main challenges to the deployment of wind energy, however, is its inherent variability and lower level of predictability than are common for other types of power plants (Lei et al., 2009; Chen and Yu, 2014; Li et al., 2018). Hybrid

electric systems that incorporate a substantial amount of wind power therefore require some degree of flexibility from other generators in the system in order to maintain the right supply/demand balance and thus ensure grid stability (Wiser et al., 2011). Failing to manage this variability leads to scheduling errors which impact grid reliability and market-based ancillary service costs (Kavasseri and Seetharaman, 2009), while potentially causing energy transportation issues in the distribution network (Salcedo-Sanz et al., 2009) and increased risks of power cuts (Li et al., 2018). This is where wind speed forecasting can play

a significant role. Incorporating high-quality wind speed forecasts, and, in return, wind power forecasts, into electric system operations gives the system more time to prepare for large fluctuations and can thereby help mitigate the aforementioned issues (Wiser et al., 2011). The variability in the short-range, particularly over the time scale of *one to six hours* is found to pose the most significant operational challenges (Wiser et al., 2011; Li et al., 2018). The development of accurate wind speed forecasts in the short-range has therefore become increasingly important.

Short-term wind speed prediction is not just a key element in the successful management of hybrid electric power systems, it is also vital in the planning for necessary shut-downs in the face of extreme weather (Chen and Yu, 2014). Most existing turbines stop producing energy when either instantaneous gust speeds or averaged wind speeds exceed a threshold of around 25 m s$^{-1}$, after which the rotation of the blades is brought to a halt and the turbine is essentially turned off (Burton et al., 2001). Using simulations of off-shore wind power in Denmark, Cutululis et al. (2012) found that loss of wind power production

during critical weather conditions can reach up to 70 % of installed capacity within an hour. Accurate forecasts of extreme wind events can therefore provide vital foresight to help prepare the electrical grid for such shutdowns as well as the duration of their downtime (Petrović and Bottasso, 2014). The prediction of extreme wind speeds poses a considerable challenge to computer science research, however, where heavy-tailed distributions such as those of wind speed (modelled according to a





type III extreme value 'Weibull' distribution) pose a serious problem to the statistical prediction of extreme values at the upper
and lower tails of the distribution. In the case of regression this problem is referred to as imbalanced regression, which we
attempt to tackle in this paper for extreme wind speed prediction in the spatio-temporal setting using an adapted convolutional
recurrent neural network (ConvRNN) model and imbalanced regression loss.

Due to the growing utilisation of wind power as a renewable energy resource, a large amount of research has focused on
the development of new and improved methods for reliable forecasting of wind speed and wind power. These methods can
be broadly divided into either physical model based methods or statistical modelling methods (Costa et al., 2008; Lei et al.,
2009; Jung and Broadwater, 2014). Physical model based methods, such as numerical weather prediction (NWP) models, are
highly capable of modelling the state of the atmosphere and have been used extensively for wind speed forecasting (see e.g.
Alessandrini et al., 2013; Deppe et al., 2013; Kikuchi et al., 2017; Cheng et al., 2017). However, due to high computational
demands they tend to have a long temporal lag (depending on their domain coverage, spatial resolution and temporal forecast
frequency) which means that for the nowcasting and short-time prediction range NWP forecasts are typically not available
on time. The physical model based methods, furthermore, suffer drawbacks due to the often laborious acquisition of the
site-specific physical data (Jung and Broadwater, 2014) and the fact that the predictive capability of NWP models degrades
significantly for highly stochastic variables like wind (Chen and Yu, 2014). In practice, physical approaches are often combined
with statistical post-processing methods into so-called hybrid physical–statistical methods in order to utilise the advantages of
both methods while mitigating the restrictions of NWP models (Chen and Yu, 2014). For examples of physical–statistical
hybrids, see e.g. Scheuerer and Hamill (2015), Dabernig et al. (2017) or Cheng et al. (2017) and references therein.

Alternatively, statistical modelling (i.e. data-driven) methods have proved to be another viable solution for the problem of
weather prediction. Among these, there has been a particularly strong trend in the past years towards deep artificial neural
networks, also termed deep learning (DL). In fact, the artificial neural network (ANN) is one of the most widely used statistical
models for wind speed and power forecasts (Jung and Broadwater, 2014), and renewable energy forecasting in general (Leva
et al., 2017). The power of the ANN lies in its ability to model highly complex and non-linear relationships between input and
output while requiring no prior assumption on the mathematical relationship between them (Jung and Broadwater, 2014). Deep
(i.e. multi-layered) ANNs are capable of automatically and effectively learning hierarchical feature representations from raw
input data, where different layers in the network essentially learn to detect different features in the data. This is different from
other physical and statistical approaches, where features are first hand-crafted from the data and then given to the model (Wang
et al., 2020). The above qualities have made deep learning models particularly attractive to the area of spatio-temporal sequence
forecasting (STSF), where complex spatial and temporal correlations are typically present in the data (Wang et al., 2020). With
the utilisation of multi-layered structures of both convolutional neural networks (CNN) and recurrent neural network (RNN)
such correlations can be learned very effectively directly from the data (Wang et al., 2020). For excellent review papers on deep
learning applications to STSF we refer the reader to Shi and Yeung (2018), Amato et al. (2020) or Wang et al. (2020).

Deep learning can be applied to STSF in myriad ways. Srivastava et al. (2015) proposed the usage of a multi-layered, fully
connected long short-term memory (FC-LSTM) network for video frame prediction by flattening the input images directly into
arrays to be used by the network. Oh et al. (2015) instead used 2D CNNs to encode the input frames before feeding them into the





LSTM network. Shi et al. (2015) improved upon these methods with the proposed convolutional LSTM (ConvLSTM) network,
embedding 2D CNNs into the LSTM network structure, which the authors applied to precipitation nowcasting. A similar
extension was made to the gated recurrent unit (GRU) network by Shi et al. (2017), the ConvGRU, which was also applied to
precipitation nowcasting, where it demonstrated a superior ability to capture rotating precipitation fields. Instead of using a 2D
CNN to capture spatial correlations only, 3D CNN models may also be used instead to perform convolution over both spatial
and temporal domains using spatio-temporal filters. Vondrick et al. (2016) applied this approach to video frame prediction
and Shi et al. (2017) demonstrated its superior performance over a 2D CNN model for precipitation nowcasting. Arguably, the
combinations of 2D CNNs with RNN networks into ConvRNNs (such as the ConvLSTM) have been met with the most success,
and have been used extensively in the literature as building blocks for DL models for STSF tasks (Shi and Yeung, 2018).
Improvements to the ConvRNN network structure have been put forward, however. Shi et al. (2017) introduced the trajectory
GRU (TrajGRU) as an improvement to the ConvGRU, where the recurrent connection structure is actively learned, while Wang
et al. (2017) proposed the Predictive RNN (PredRNN) as an improvement to the ConvLSTM network by maintaining a global
memory state rather than constraining memory states to each ConvLSTM module individually. PredRNN++ was later proposed
by Wang et al. (2018), where more nonlinearities were added to the updating process of the global memory state and the authors
demonstrate the model to be superior to TrajGRU and ConvLSTM for video frame prediction. A different approach was taken
by Rao et al. (2020), where two novel spatio-temporal DL methods are proposed based on functional neural networks (FNN) as
possible improvements to the ConvRNN approaches. Generative adversarial networks (GANs) can offer yet another alternative
to STSF, a thorough review of which is provided by Gao et al. (2020).

More recently, Rasp et al. (2020) created a benchmark data set for data-driven spatio-temporal forecasts which has been used
extensively since its publication. Rasp and Thuerey (2021) used a ResNet to predict three parameters (geopotential, temperature
and precipitation) with a coarse spatial resolution of 5.625○ for up to five days ahead whereas Weyn et al. (2020) used a
convolution neural network on a cubed sphere. A different approach was followed by Pathak et al. (2022) who used a Fourier-
based neural network for forecasting surface wind speed and total precipitation on a global scale, with a spatial resolution of
0.25○ and for lead times of up to ten days. Lastly, Keisler (2022) implemented a graph neural network based approach for
prediction of 500 hPa geopotential and 850 hPa temperature, transforming the gridded information on an iconohexadral grid
and back to a latitude-longitude grid as output and were able to achieve good results for the first days.

Deep learning has been used also in the context of extreme weather forecasting. Liu et al. (2016) developed a multi-channel
CNN model to classify images of extreme weather events such as tropical cyclones, atmospheric rivers and weather fronts.
Racah et al. (2017) followed a similar approach but utilised a multi-channel 3D CNN architecture to classify extreme weather
events spatially as well as temporally. Feng and Fox (2021) proposed the TSEQPredictor model for earthquake prediction
over Southern California, which combines a CNN autoencoder with a temporal convolutional network (TCN) to classify the
occurrences extreme earthquake events. The authors were able to improve their model by employing skip connections and
local temporal attention into the network. Yu et al. (2017), on the other hand, proposed modelling spatial extreme events
by bridging a gap between traditional statistical methods and graph methods via decision trees, while Thomas et al. (2021)



employed an unsupervised k-means clustering approach to investigate weather patterns responsible for extreme wind speed events throughout Mexico.

The definition of the term 'extreme event' can vary substantially in the spatio-temporal context, however. In the literature, extreme events often refer to hazardous weather *patterns*, present over some spatial or spatio-temporal domain. While these events are certainly extreme within the underlying climatology of the study, they are not usually extreme with respect to the data distribution used for the study. For example, many classification studies of extreme weather patterns ensure that the model is supplied with an equal number of negative and positive samples, so as to avoid any model biases due to class imbalances.

Even when class imbalances are tolerated, there are other remedies available such as resampling strategies (see: e.g. Oliveira et al., 2021, for a spatio-temporal approach), one-class classification approaches (e.g. Deng et al., 2018; Goyal et al., 2020) or deep anomaly detection (e.g. Hendrycks et al., 2019). Extreme events in regression problems, on the other hand, typically refer to the tail of the data distribution i.e. highly underrepresented values in the data that are therefore rarely encountered during model training. Regression problems on imbalanced data distributions are termed imbalanced regression problems. Ding et al.

(2019) provide a formal analysis on why DL regression models suffer from overfitting and underfitting problems when data is imbalanced and propose a novel loss function called the extreme value loss (EVL), based on extreme value theory, which is demonstrated to improve predictions on extreme events in time-series forecasting. The authors furthermore propose a memory network based neural network architecture to memorise past extreme events for better prediction in the future. Ribeiro and Moniz (2020) addressed the problem of imbalanced regression by proposing the squared error-relevance area (SERA) loss

function, based on the idea of 'relevance functions'. Yang et al. (2021), on the other hand, proposed the idea of distribution smoothing to address underrepresented or even missing labels in the label distribution and reduce unexpected similarities within the feature distribution that arise due to the label imbalance. The smoothed label distribution can then be used easily for re-weighting methods, where the loss function can be weighted by multiplying it with the inverse of the smoothed label distribution for each target. Such re-weighting of the loss function is a cost-sensitive remedy to data imbalance and has been

used in the context of spatio-temporal weather forecasting, for example by Shi et al. (2017) for precipitation nowcasting.

   In this paper, we propose a deep convolutional LSTM (ConvLSTM) model for extreme wind speed prediction, adapted with imbalanced regression loss to account for the heavy tails. To this end, we investigate the inversely weighted mean absolute error (W-MAE), the inversely weighted mean squared error (W-MSE) and the squared error relevance area (SERA) loss functions. The performance of our adapted model is compared against the standard mean absolute error (MAE) and mean squared error

(MSE) losses, as determined from a spatio-temporal forecast verification using the symmetric extremal dependency index (SEDI).

## 2   Methodology

### 2.1   Data Collection and Preprocessing

The wind speed data used in this work was downloaded from the Copernicus Climate Change Service Climate Data Store
(CDS) of the ECMWF (see Hersbach et al., 2018). The reanalysis data of the U and V components of the horizontal wind



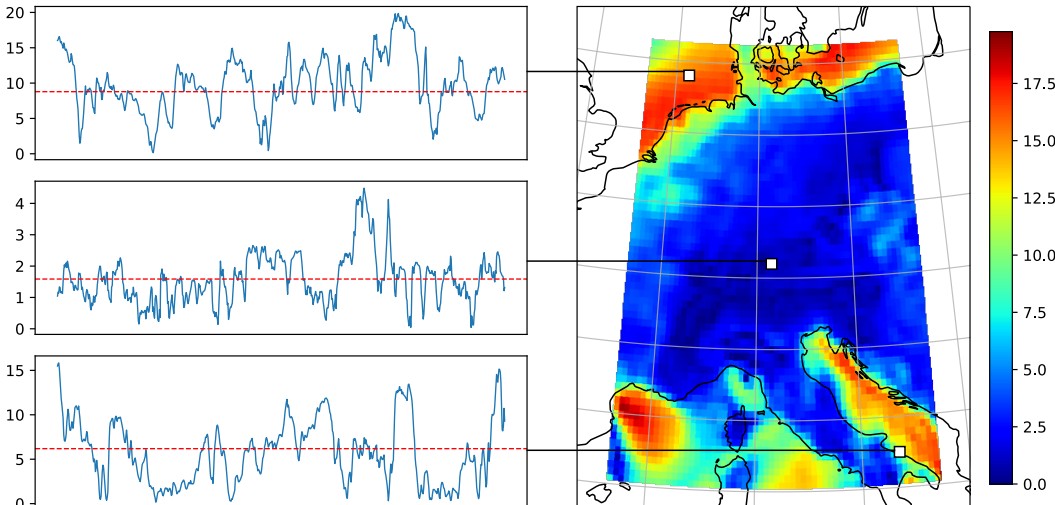

**Figure 1.** A visualisation of the wind speed data (in m s$^{-1}$). The right figure shows a color-map of an example data frame, overlaid on a cartographic map (Central Europe) showing the coastlines of the region. On the left, the wind speed time series of three arbitrary locations (white squares) within the frame are plotted for the duration of one month, as well as the climatological means at these locations (dotted red lines).

velocity (in m s$^{-1}$) were taken at 1000 hPa from the *ERA5 hourly data on pressure levels from 1979 to present* dataset. By computing the square root of the sum of the squares of the two wind velocity components the scalar wind speed was obtained. The data was collected with a temporal resolution of one hour between 01 January 1979 and 01 January 2021 (42 years) on a spatial grid over central Europe. Of these data, the last two years between 2019-2021 were held out as testset. The eight years between 2011-2019 were used for training and validation in the first part of the experiment, dedicated to model optimisation. In the second part of the experiment the optimal models were then trained and validated on the entire 40 years between 1979-2019, using the years between 2017-2019 as validation.

The spatial grid comprises $64 \times 64$ grid points between 40–56° N and 3–19° E, the spatial resolution being 0.25° ($\approx 28$ km). This region was selected for its geographical variation, as it includes both land and sea regions as well as flat and mountainous areas, while the region is furthermore divided into different climatic regions such as the Pannonian climate region in Eastern Austria and the Alpine climate region covering the Austrian Alpine range. Interplay between these features can result in highly complex wind dynamics, which is where we expect the application of deep learning to be particularly promising. Moreover, we expect the fine spatial resolution of 0.25° to be critical to capturing the complex fine-scale dynamics of a variable like low-level wind, and thus improving forecasting ability, while the resolution also marks an important step forward for data-driven models to be truly competitive with state-of-the-art numerical weather prediction models, which are run at $\approx 0.1°$ resolution (Pathak et al., 2022).



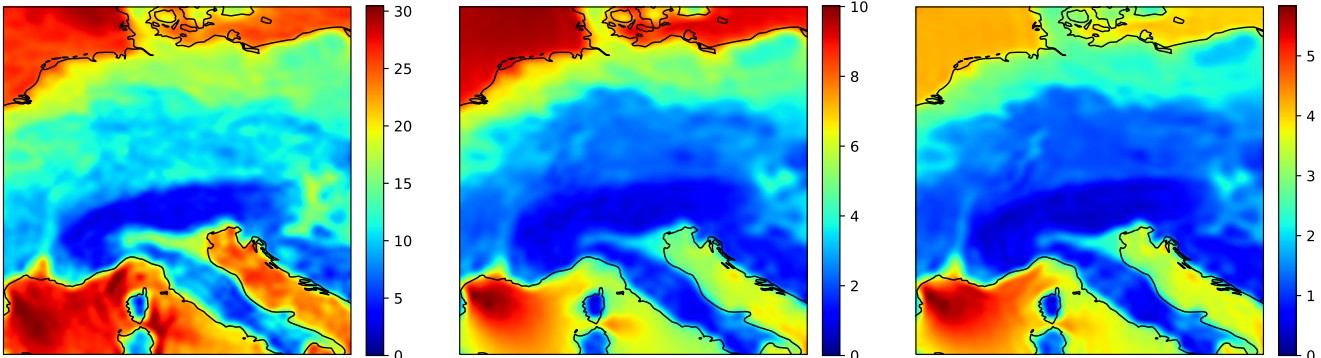

**Figure 2.** Color-maps of the maximum (left), mean (center) and standard deviation (right) of the wind speeds (in m s$^{-1}$) over the region. The figures display a sharp division of the statistics along the coastlines.

A visualisation of the data is provided in Fig. 1. The figure shows on the right an example time slice and on the left wind speed time series of three arbitrary locations over the duration of one month, including the climatological means at these locations. Evidently the local climatological means (and by extension, the local wind speed distributions) vary substantially throughout the region, where striking differences in magnitude can be seen between the off-shore and on-shore regions. To highlight these spatial differences, Fig. 2 shows the maximum, mean and standard deviation of the wind speed over the region, which unveil a sharp division of the statistics with the underlying coastlines of the region. Indeed, extreme winds (e.g. larger than 25 m s$^{-1}$) seem to occur almost exclusively off-shore. If there were, in fact, stronger winds present over this region of mainland Europe between 1979 and 2021 then they have not been captured by the hourly ERA5 reanalysis.

Rather than defining extreme winds in terms of their absolute severity, we proceed to define extreme winds, instead, in terms of their *relative rarity* at each coordinate. This definition focuses the forecasting problem on the tails of the respective distributions at each coordinate, which ensures that the forecasting of extremes is conducted over the entire region, rather than only locally over some particularly dominant area. By selecting a distributional percentile (e.g. the 99th percentile), we then define extreme winds as those wind speeds surpassing this percentile threshold of the wind speed distribution at the respective coordinate. To this end, the raw wind speed data were standardised with a local Z-normalisation at each coordinate, which centers each local distribution around zero mean with unit standard deviation according to the following transformation:

$$\mathbf{x}'_{i,j} = \frac{\mathbf{x}_{i,j} - \mu_{i,j}}{\sigma_{i,j}} \tag{1}$$

where $\mathbf{x}'_{i,j}$ denotes the transformed variable, $\mathbf{x}_{i,j}$ the original variable and $\mu_{i,j}$ and $\sigma_{i,j}$ the mean and standard deviation, respectively, at the coordinate $i,j$. The Z-normalised data thus represent the wind speed in terms of the number of standard deviations from the respective mean at each coordinate.





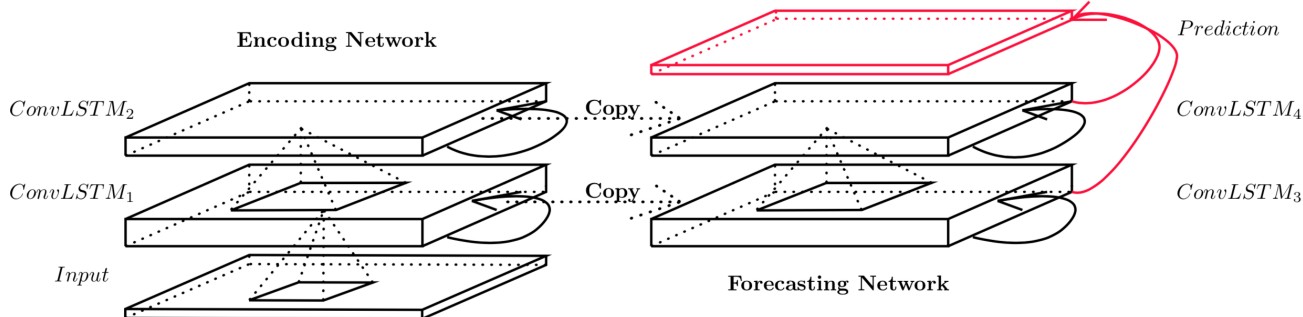

**Figure 3.** The multi-layered encoding–forecasting ConvLSTM network. The hidden states and cell outputs of the encoding network are copied to the forecasting network, from which the final prediction is made. **© Shi et al. (2015). Used with permission.**

## 2.2 Model Description

The model implemented and adapted for the task of spatio-temporal forecasting of extreme events is an adaptation of the convolutional long short-term memory (ConvLSTM) network, as proposed by Shi et al. (2015) for precipitation nowcasting. However, while Shi et al. (2015) trained their ConvLSTM model using cross-entropy loss, we propose adapting the model to
forecasting of extreme events by utilising two types of loss functions from the literature on imbalanced regression: Weighted loss and the squared error-relevance area (SERA) loss.

The ConvLSTM is an example of a ConvRNN model, which forms a synthesis of a convolutional neural network (CNN) and a recurrent neural network (RNN). CNNs are a class of feedforward artificial neural networks, used primarily for data mining tasks involving spatial data and have gained a lot of attention in the area of computer vision and natural language
processing (Ghosh et al., 2020), while RNNs are known for their powerful ability to effectively model temporal dependencies (Shi et al., 2015). By utilising the strengths of the CNN to capture spatial correlations and the RNN to capture temporal correlations in the data, ConvRNN models have demonstrated promising forecasting ability in the spatio-temporal setting. As a deep ConvRNN model, the ConvLSTM has the potential to effectively model the complex dynamics of the spatio-temporal wind speed forecasting problem.
We adopt the deep ConvLSTM model with an encoding–forecasting network structure, as is common for spatio-temporal sequence forecasting, where both encoding and forecasting networks consist of several stacked ConvLSTM layers. As depicted in Fig. 3, the encoding ConvLSTM network compresses the input into a hidden state tensor and the forecasting ConvLSTM network unfolds this hidden state into the final prediction. We implement the model as a multi-frame forecasting model, with 12 hour input and 12 hour prediction.

### 2.2.1 Inversely Weighted Loss

In order to combat the effects of data imbalance on the imbalanced regression problem, we adapt the ConvLSTM model with two different loss functions that have been proposed for imbalanced regression problems. The first of these is the relatively





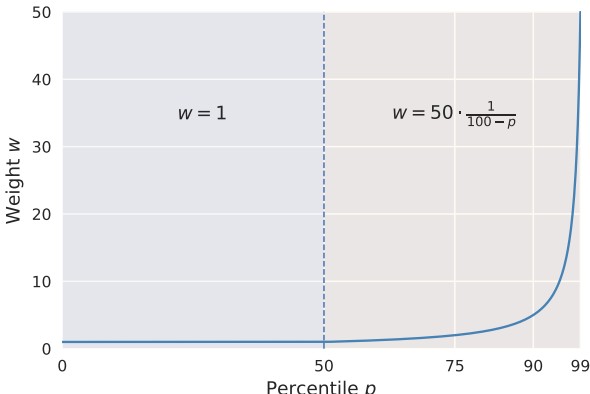

**Figure 4.** Weighting function $w$ used to construct the inversely weighted mean squared error (W-MSE) and inversely weighted mean absolute error (W-MAE).

simple weighted loss, which consists of assigning a weight $w(y)$ to each value in the input frame according to its target wind speed $y$. For a loss function $L$ of the target $y$ and prediction $\hat{y}$ (consisting of $N$ time-frames of $M \times M$ spatial coordinates) and a weighting function $w(y)$, the weighted loss $L_W$ is computed as:

$$L_W(\hat{y}, y) = \frac{1}{N} \sum_{n=1}^{N} \sum_{i,j=1}^{M} w(y_{n,i,j}) \cdot L(\hat{y}_{n,i,j}, y_{n,i,j}) \tag{2}$$

As weighted loss functions we investigate both the weighted mean squared error (W-MSE) loss and the weighted mean absolute error (W-MAE) loss. We proceed to compute the weights in proportion to the inverse of of the data distribution for each target, as suggested by Yang et al. (2021). For a continuous target distribution, this typically implies discretising the distribution into intervals (see e.g. Shi et al., 2017), where all predictions within an interval are weighted by the same weight. Due to our definition of extreme events in terms of local percentile thresholds we proceed to discretise the target distribution into intervals spanning the percentage of the distribution between percentile $p$ and 100. For a set of increasing percentiles $\mathcal{P} = \{p_k\}$, all targets $p_k \leq y < p_{k+1}$ are then weighted proportionally to the inverse of the percentage between $p_k$ and 100 i.e. $w(y) \propto 1/(100 - p_k)$. We utilise a range of integer percentiles $\mathcal{P} = \{p_k | k \in [50, 99]\}$ and normalise weights such that the interval between percentiles 50 and 51 is given unit weight. All values smaller than the 50th percentile ($p_{50}$) are also given unit weight. This results in the weighting function shown in Eq. 3, which is also presented graphically in Fig. 4.

$$w(y) = \begin{cases} 1 & \text{if} \quad y < p_{50} \\ 50 \cdot \frac{1}{100 - p_k} & \text{if} \quad p_k \leq y < p_{k+1} \quad \text{for} \quad k \in [50, 99] \end{cases} \tag{3}$$





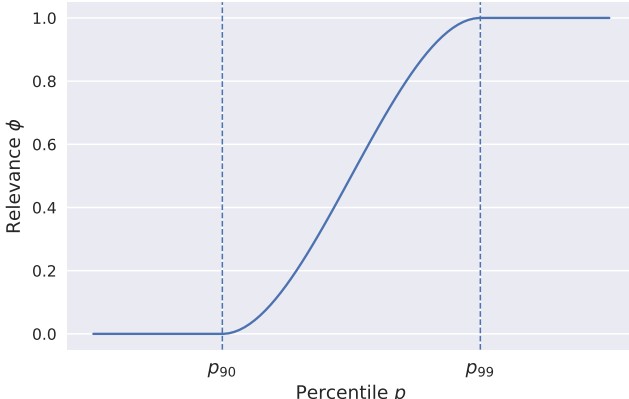

**Figure 5.** The relevance function $\phi$ obtained by interpolating the 90th percentile ($p_{90}$) as control-point of minimum relevance and the 99th percentile ($p_{99}$) as control-point of maximum relevance, using the *pchip* algorithm of Ribeiro and Moniz (2020).

### 2.2.2 Squared Error-Relevance Area Loss

As a second approach to combating data imbalance, we investigate the squared error-relevance area (SERA) loss, as proposed

by Ribeiro and Moniz (2020). The SERA loss is based on the concept of a *relevance function* $\phi : \mathcal{Y} \longrightarrow [0, 1]$, which maps the target variable domain $\mathcal{Y}$ onto a $[0, 1]$ scale of relevance. The relevance function $\phi$ is determined through a cubic Hermite polynomial interpolation of a set of 'control-points'. The set of control-points $S = \{\langle y_k, \phi(y_k), \phi'(y_k)\rangle\}_{k=1}^s$ are user-defined points where the relevance may be specified, which are typically local minima or maxima of relevance and thus all have derivative $\phi'(y_k) = 0$ (Ribeiro and Moniz, 2020). We define the 90th percentile ($p_{90}$) of the standardised wind speed distribution

at each coordinate as the point of minimum relevance at that coordinate $\langle y_1 = p_{90}, \ \phi(y_1) = 0.0, \ \phi'(y_1) = 0.0\rangle$ and the 99th percentile ($p_{99}$) as the point of maximum relevance $\langle y_1 = p_{99}, \ \phi(y_1) = 1.0, \ \phi'(y_1) = 0.0\rangle$. The interpolation is carried out according to Ribeiro and Moniz (2020) by using the piecewise cubic Hermite interpolating polynomials (*pchip*) algorithm and the obtained relevance function is shown in Fig. 5.

Defining $D^t$ as the subset of data pairs for which the relevance of the target value is greater or equal than a cut-off $t$, i.e.

$D^t = \{\langle x_i, y_i\rangle \in D | \phi(y_i) \geq t\}$, the squared error-relevance $SER_t$ of the model with respect to the cut-off $t$ is computed as follows:

$$SER_t = \sum_{i \in D^t} (\hat{y}_i - y_i)^2 \tag{4}$$

where $\hat{y}_i$ and $y_i$ are the $i$'th prediction and target values, respectively. The curve obtained by plotting $SER_t$ against $t$ is decreasing and monotonic (Ribeiro and Moniz, 2020) and provides an overview of how the magnitudes of the prediction errors

change on subsets comprising varying degrees of relevant samples ($t = 0$ representing all samples and $t = 1$ representing only the most relevant samples). Finally, the squared error-relevance area (SERA) is defined as the area under the $SER_t$ curve:





**Table 1.** The number of parameters of the ConvLSTM model with different numbers of layers.

| ConvLSTM layers | Number of parameters |
|:---:|:---:|
| 2 | 2,385,953 |
| 3 | 10,061,025 |
| 4 | 34,201,185 |
| 5 | 62,060,641 |

$$SERA = \int\limits_{0}^{1} SER_t \, dt \tag{5}$$

The smaller the area under the curve is, the better the model is. We note that assigning uniform relevance values to all data points recovers the MSE loss.

### 2.2.3 Implementation

The model was implemented and trained using Pytorch, the code of which can be found under: https://github.com/dscheepens/Deep-RNN-for-extreme-wind-speed-prediction. In addition to the different loss functions, we investigate model architectures with different numbers of ConvLSTM layers, ranging from 2–5 layers (in both the encoder and the forecasting networks). The numbers of parameters of all model architectures are shown in Table 1. In line with Shi et al. (2015), all layers utilise $3 \times 3$ kernels. The convolution over each successive filter operates such as to successively halve the spatial dimensions of the input, while the number of hidden states (features) are successively doubled (starting from 16 hidden states).

We trained our models using mini-batch gradient descent with a batch-size of 16 and used the adaptive moment estimation (Adam) as optimiser. Adam optimiser is a popular and reliable choice for deep learning neural networks which computes adaptive learning rates for each parameter of the model, based on their update frequency (see e.g. Ruder, 2017). The initial learning rate of the optimiser was set to $10^{-3}$. During training, early-stopping was performed on the validation set to ensure that the model with the lowest validation loss was saved as the best model and thus to avoid overfitting the model. The early-stopping mechanism was set up to stop training when the validation loss failed to decrease for 20 consecutive epochs.

### 2.3 Verification

Since the ConvLSTM model investigated in this study is a spatio-temporal forecasting model, it appears in order to evaluate the model with a verification method that captures forecasting ability at different temporal, as well as spatial scales. In order to evaluate the model at different spatial scales we utilise the minimum coverage method, as proposed by Damrath (2004). As a filtering method, the minimum coverage method works well for verifying 'messy' forecasts that do not contain well-defined features (Ebert, 2009), which we expect to be particularly applicable to wind speed due to its highly stochastic behaviour.





Another advantage of the method is that it is parameter-free and easy to implement. While the method is a spatial forecast
verification method, it can be applied in a simple manner *per* lead-time for a temporal assessment.

The minimum coverage method essentially states that "a forecast is useful if the event is predicted over a minimum fraction
of the region of interest" (Ebert, 2008). Denoting $\langle P \rangle_s = \frac{1}{n} \sum_n I$ to be the fraction of grid points with events $I \in \{0,1\}$ within
a neighbourhood of scale $s$, the entire neighbourhood is classified as the event $\langle I \rangle_s$ according to:

$$\langle I \rangle_s = \begin{cases} 0 & \langle P \rangle_s < P_e \\ 1 & \langle P \rangle_s \geq P_e \end{cases} \qquad (6)$$

where $P_e$ is the minimum fraction of the neighbourhood that must be covered by events in order for the neighbourhood to be
classified as an event. A neighbourhood of scale $s$ refers to a squared area of dimension $s \times s$ grid-points. Due to the scarcity
of extreme events in the data, we chose to use a minimum coverage criterion with $P_e$ set to the value $1/n$ i.e. we require only a
single event to be present in the neighbourhood for the neighbourhood to be classified as an event. The neighbourhood events
can then be evaluated from a typical $2 \times 2$ contingency table using any desired categorical score. The categorical score used
here is the symmetric extremal dependence index (SEDI), which is computed as follows:

$$\text{SEDI} = \frac{\log F - \log H + \log(1-H) - \log(1-F)}{\log F + \log H + \log(1-H) + \log(1-F)} \qquad \in [-1,1] \qquad (7)$$

where $H$ and $F$ are the hit rate and false alarm rate, respectively. The SEDI was chosen for its unique property of non-
degeneracy for rare events. Stephenson et al. (2008) have shown that practically all categorical scores degenerate to trivial
values such as 0, 1 or infinity for exceedingly rare events, i.e. as the base rate of the event tends to zero. The SEDI was proposed
by Ferro and Stephenson (2011) as a remedy to the degeneracy problem and, in fact, combines more desirable properties into
one score than any other categorical score (Hogan and Mason, 2012, p. 54).

As is typical for neighbourhood methods, scores were computed for a set of scales and a set of intensity thresholds in order to
provide a diagnostic assessment of forecast quality on spatial scale and intensity (see: e.g. Ebert, 2009). As such, we computed
the SEDI for a set of scales corresponding to approx. 28, 83, 139, 194 and 250 km, and a set of thresholds corresponding to
the local 50th, 75th, 90th, 95th, 99th and 99.9th percentiles of the standardised wind speed distribution at each coordinate: We
remind the reader that the models are judged on their ability to forecast extreme events in terms of *relative rarity*, which we
measure as an event's percentile with respect to the local frequency distribution at the respective coordinate. Finally, in order
to obtain an aggregated result over all forecasts made by a model, the elements in the contingency table are aggregated over all
forecasts and the scores are computed subsequently from the aggregated contingency table.
In the next section we present the results obtained from combining the multi-layered ConvLSTM network with inversely
weighted mean absolute error (W-MAE), inversely weighted mean squared error (W-MSE) and squared error-relevance area
(SERA) loss and compare these against the standard mean absolute error (MAE) and mean squared error (MSE) loss. The
optimal number of layers for each model is determined from the minimum validation-loss obtained by the networks over the




4-fold cross validation process, as conducted over the 8 years of data between 2011–2019. The optimal models are then re-

trained using the entire 40 years of data between 1979–2019 (using 2017-2019 as validation) and their results are compared on

the held-out test set (comprising the years 2019-2021) using the SEDI. Finally, the best performing model is further analysed

with the minimum coverage method and its forecasts are visualised.

## 3  Results

We begin by showing, in Table 2, the minimum validation-loss obtained by the ConvLSTM network with number of layers

ranging between 2–5 to determine the optimal number of layers for each loss function, which is emphasised in boldface. As

described in the methodology, the models were then trained once again using the full 40 years of data and the optimal number

of network layers for each loss function. These optimal models are compared in Table 3 over local intensity thresholds varying

between the 50th and 99.9th percentiles, using the symmetric extremal dependency index (SEDI). The optimal number of net-

work layers used with each model is shown in brackets next to the name of the loss function. It is evident from Table 3 that the

usage of imbalanced regression loss results in superior SEDI scores for extreme events between the 75th and 99th percentiles.

While outperformed by the SERA loss, the W-MAE and W-MSE also show clear improvements over the standard MAE and

MSE loss from the 90th percentile onward. Indeed, the table shows how the usage of imbalanced regression loss manages to

shift optimal performance towards the extreme intensity thresholds, as opposed to performance simply decreasing monotoni-

cally for increasingly rare events, as is the case for the standard MAE and MSE from the 75th percentile onward. In the table

are included, for reference, the SEDI scores achieved by a simple persistence forecast, which is a forecast consisting simply

of a repetition of the final observation (input) frame. It is clear that the improvement offered by the imbalanced regression loss

functions ceases around the 99.9th percentile threshold, where SEDI scores are comparable among all models, while being, in

addition, only marginally better than persistence.

**Table 2.** Minimum validation loss as obtained by the ConvLSTM network with number of layers ranging from 2–5 (denoted in brackets) and trained with either W-MAE, W-MSE, SERA, MSE or MAE loss. Values are presented as the mean $\pm$ one standard deviation from the 4-fold cross-validation. The optimal model for each loss function is emphasised in boldface.

|  | ConvLSTM (2) | ConvLSTM (3) | ConvLSTM (4) | ConvLSTM (5) |
|---|---|---|---|---|
| W-MAE | $(8.3 \pm 0.6) \cdot 10^{-2}$ | $(8.2 \pm 0.6) \cdot 10^{-2}$ | $(8.2 \pm 0.6) \cdot 10^{-2}$ | $\mathbf{(8.1 \pm 0.5) \cdot 10^{-2}}$ |
| W-MSE | $(8.2 \pm 0.7) \cdot 10^{-2}$ | $(8.1 \pm 0.6) \cdot 10^{-2}$ | $(8.0 \pm 0.7) \cdot 10^{-2}$ | $\mathbf{(7.8 \pm 0.7) \cdot 10^{-2}}$ |
| SERA | $(24.5 \pm 0.8) \cdot 10^{-3}$ | $(24.2 \pm 1.0) \cdot 10^{-3}$ | $(24.0 \pm 0.8) \cdot 10^{-3}$ | $\mathbf{(23.9 \pm 1.1) \cdot 10^{-3}}$ |
| MAE | $(24.3 \pm 0.4) \cdot 10^{-3}$ | $(24.0 \pm 0.5) \cdot 10^{-3}$ | $\mathbf{(23.6 \pm 0.4) \cdot 10^{-3}}$ | $(23.7 \pm 0.4) \cdot 10^{-3}$ |
| MSE | $(18.6 \pm 0.6) \cdot 10^{-3}$ | $(18.3 \pm 0.5) \cdot 10^{-3}$ | $(17.9 \pm 0.6) \cdot 10^{-3}$ | $\mathbf{(17.6 \pm 0.4) \cdot 10^{-3}}$ |

We investigate the performance of these models further in Fig. 6, where the SEDI scores obtained by each model are plotted

per lead-time (in hours) for the 99th percentile intensity threshold. We, once again, include in this comparison the persistence

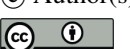



**Table 3.** Comparison of SEDI scores obtained by the ConvLSTM network trained with either W-MAE, W-MSE, SERA, MAE or MSE loss, presented for winds ($y$) exceeding local intensity thresholds varying between the 50th and 99.9th percentiles. The optimal number of network layers used for each loss function is given in brackets after the name of the loss function. The persistence forecast is included in the table for reference. The table shows that the usage of imbalanced regression loss allows to substantially improve forecasts of local wind speeds exceeding the 75th percentile threshold.

| | $y \geq p_{50}$ | $y \geq p_{75}$ | $y \geq p_{90}$ | $y \geq p_{95}$ | $y \geq p_{99}$ | $y \geq p_{99.9}$ |
|---|---|---|---|---|---|---|
| W-MAE (5) | 0.828 | **0.871** | 0.885 | 0.881 | 0.839 | **0.734** |
| W-MSE (5) | 0.801 | 0.862 | 0.885 | 0.885 | 0.828 | 0.704 |
| SERA (5) | 0.767 | 0.832 | **0.893** | **0.921** | **0.850** | 0.719 |
| MAE (4) | **0.854** | 0.867 | 0.847 | 0.828 | 0.784 | 0.725 |
| MSE (5) | 0.848 | 0.861 | 0.844 | 0.824 | 0.783 | 0.712 |
| Persistence | 0.689 | 0.729 | 0.732 | 0.731 | 0.724 | 0.707 |

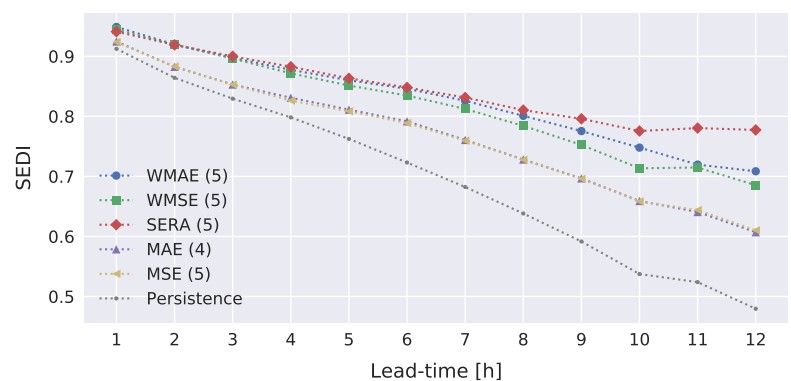

**Figure 6.** Comparison of SEDI scores obtained by the ConvLSTM network trained with either W-MAE, W-MSE, SERA, MAE or MSE loss, plotted over lead-time (in hours), for local extreme events of the 99th percentile intensity threshold. The optimal number of network layers used for each loss function is given in brackets after the name of the loss function. The label 'persistence' refers to the persistence forecast. The comparison shows that the superior scores obtained by the SERA loss in Table 3 are due in particular to its better performance on lead-times 6–12 hours.

forecast for reference. The figure shows that the superior performance of the SERA loss over the W-MAE and W-MSE in Table 3 results from improved performance on lead-times beyond ca. eight hours; For lead-times below ca. eight hours performance of the W-MAE and W-MSE loss are very competitive with the SERA. ConvLSTM trained with standard MAE and MSE loss is certainly more informative that the persistence forecast, performance is inferior to all three imbalanced regression losses on all lead-times.



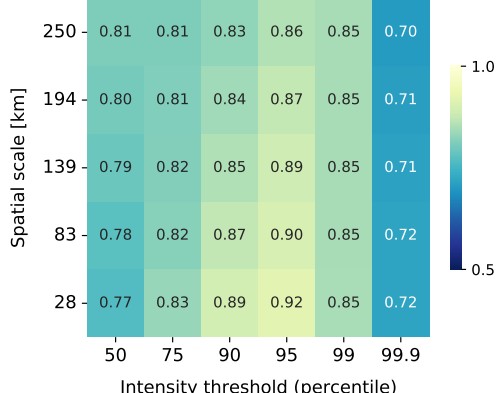

**Figure 7.** Intensity-scale diagram of SEDI scores obtained by the 5-layered ConvLSTM network trained with squared error-relevance area (SERA) loss.

In addition, in order to establish a spatial picture of forecast quality, we provide in Fig. 7 an intensity-scale diagram (see e.g. Casati et al., 2004; Ebert, 2008) of the SERA (5) model, which we highlight here due to its superior SEDI scores on intensity thresholds between the 90th and 99th percentiles in Table 3. The figure shows how the SEDI scores change both with varying intensity threshold as well as spatial scale. However, contrary to the expected behaviour of forecasting performance improving

with increasingly coarser scale, SEDI scores in the diagram in fact decrease with increasing scale (with the sole exception of the scores obtained with the 50th percentile threshold).

Lastly, we proceed to show two visualisations of the forecasts made by the different ConvLSTM models investigated in this paper, which serve to highlight their respective strengths and weaknesses. Figure 8 shows a target observation of a growing intensification of anomalous winds in the left of the frame, as well as the forecasts made by the respective models. This example

highlights the striking difference between the utilisation of imbalanced regression loss versus the standard MAE and MSE loss - both of which fail to capture the intensity of the target extremes, although they do manage to capture the general pattern of the target observation. Among the imbalanced regression losses, the SERA loss provides a substantially coarser forecast of the extreme region than the W-MAE and W-MSE, and, as such, allows for more false alarms in order to capture more of the event. This strategy can be clearly distinguished as well in Fig. 9, where the SERA-trained model severely overshoots its forecast

to capture what is only a very minor event in the target observation. Indeed, the SERA loss appears to forecast something of a coarse-grained worst-case scenario, while the W-MAE and W-MSE forecasts are sharper and more conservative. These opposing characteristics lead to a strong suspicion that an ensemble of all three models (forecasting the average of the forecasts made by the models) may be worthwhile investigating further. These ensemble forecasts are included in Fig. 8 and Fig. 9 in the bottom row. While the ensemble forecast in Fig. 8 shows that some of the extreme intensities captured with the SERA

loss are lost in averaging process, the ensemble forecast is significantly sharper spatially and continues to provide a substantial



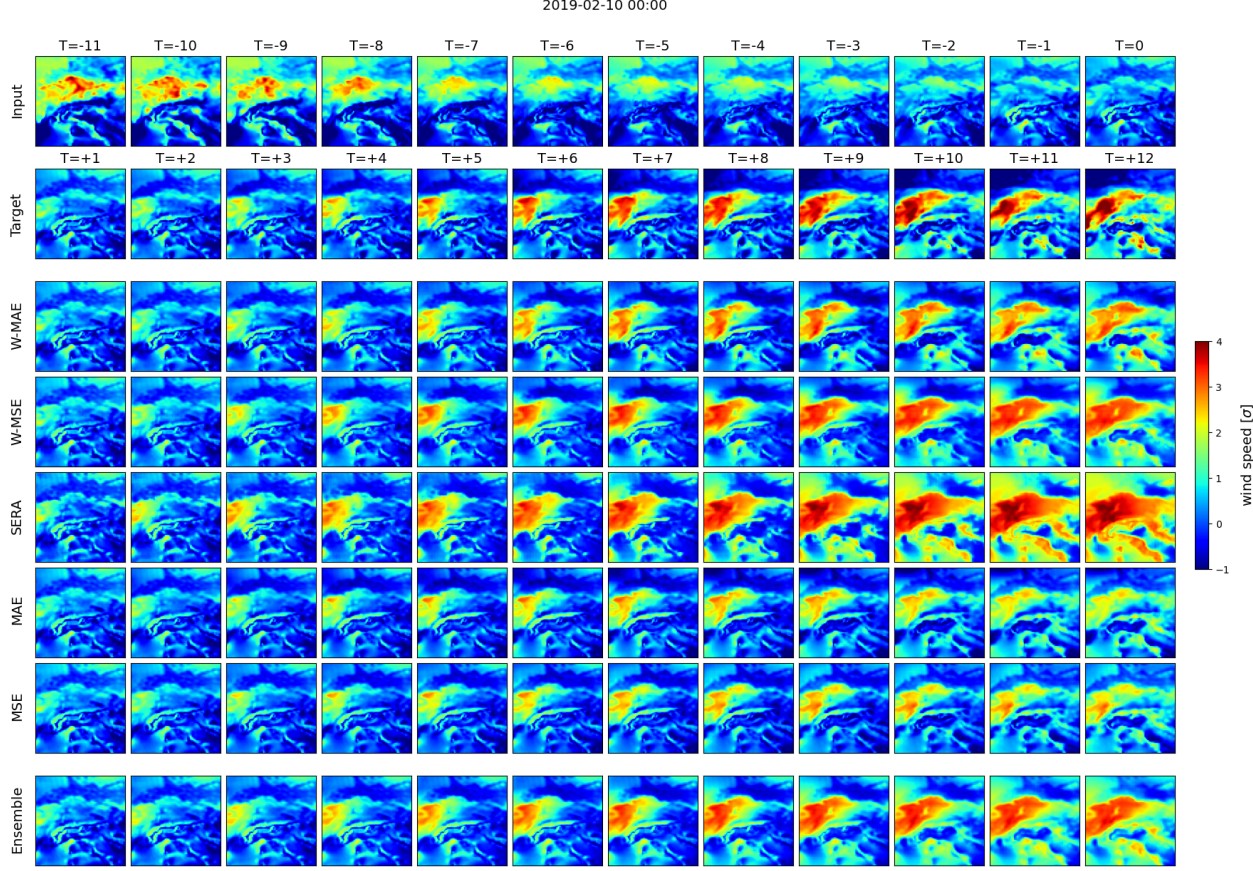

**Figure 8.** An example forecast from the different ConvLSTM networks trained with either W-MAE, W-MSE, SERA or standard MAE or MSE loss. The first row from the top displays the 12 input frames, the second row the succeeding 12 target frames and the following rows the 12 predicted frames of the models. $T$ refers to the index of the frame (in hours), with $T = 0$ denoting the last input frame and $T = +12$ denoting the final target and prediction frames. The final row shows the averaged forecast of an ensemble of the W-MAE, W-MSE and SERA-trained models.

improvement over the MAE and MSE. The ensemble forecast in Fig. 9, furthermore, shows how the overshooting of the SERA-trained model is significantly limited and large swaths of false alarms avoided.

A selection of further forecast visualisations can be found in the supplements, or in our GitHub repository: https://github.com/dscheepens/Deep-RNN-for-extreme-wind-speed-prediction.

Another way to highlight the differences in forecasts between the different models is to look at frequency bias, which is presented in Table 4 for the same set of intensity thresholds as before. The tendency of the SERA-trained model to severely overshoot the target observation with large swaths of false alarms is reflected in Table 4 by substantially increased frequency





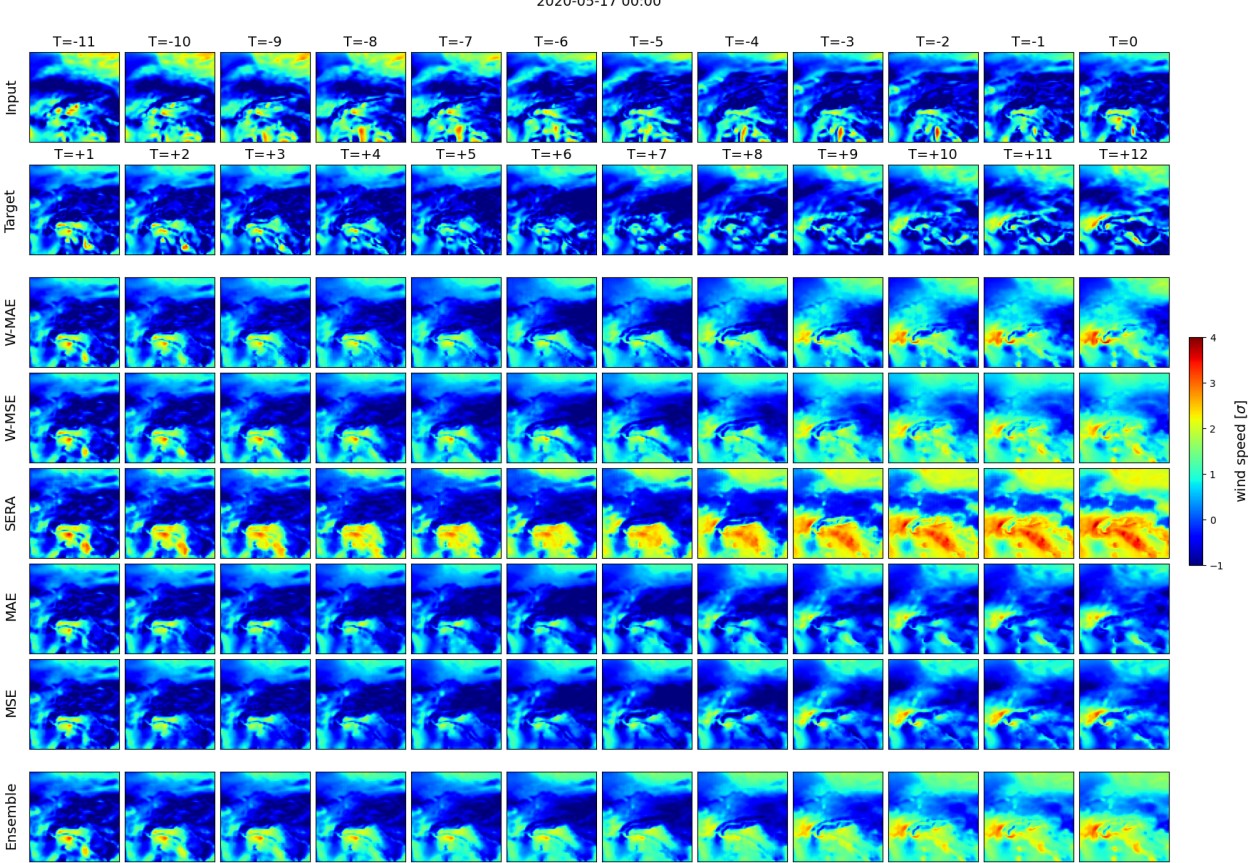

**Figure 9.** As Fig. 8

bias for extreme events between the 75th and 99th percentiles, as compared with the other models. While the W-MAE and W-MSE do present higher frequency bias than their unweighted counterparts, these differences are comparatively minor. As

expected, the ensemble forecast significantly limits the frequency bias for all events, as the different biases of the models counteract one another. The low frequency bias of all models in forecasting extreme events of the 99.9th percentile threshold offers another clue as to why the SEDI scores of the models for such events are so poor: these events are less frequently forecasted at all and are thus more often missed.

     Given its continued widespread usage, we present in Table 4 also the root mean squared error (RMSE) obtained by the

different models, as aggregated over all forecasts in the testset. The substantial frequency bias of the SERA-trained model in its forecasting of extreme events is reflected here by a substantially increased RMSE, as errors between extremes and non-extremes occur more often due to common misplacement of extreme event forecasts. The RMSE of the ensemble model highlights the effectiveness of the ensemble in limiting these large-scale errors produced by the SERA-trained model. Although




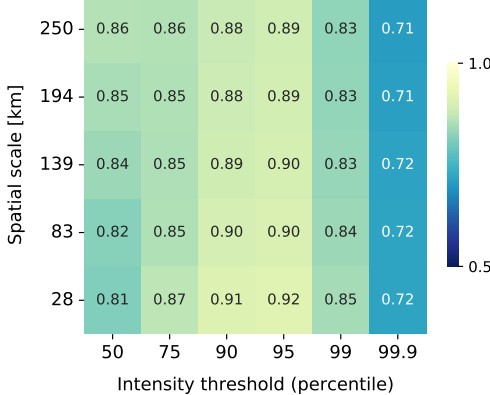

**Figure 10.** Intensity-scale diagram of SEDI scores obtained by the ensemble model consisting of the W-MAE, W-MSE and SERA-trained
ConvLSTM networks.

the aggregated RMSE scores tell us nothing about the quality of forecasts of extreme events, they do give us an overall picture
of the typical magnitude of errors made by the models.

**Table 4.** Comparison of frequency bias (in %) and RMSE of the ConvLSTM network trained with either W-MAE, W-MSE, SERA, MAE or
MSE loss. Frequency bias is presented for winds ($y$) exceeding local intensity thresholds varying between the 50th and 99.9th percentiles.
The optimal number of network layers used for each loss function is given in brackets after the name of the loss function. Also included in
the table is the ensemble model, consisting of the W-MAE, W-MSE and SERA-trained networks.

| | | | Frequency bias | | | | |
| --- | --- | --- | --- | --- | --- | --- | --- |
| | $y \geq p_{50}$ | $y \geq p_{75}$ | $y \geq p_{90}$ | $y \geq p_{95}$ | $y \geq p_{99}$ | $y \geq p_{99.9}$ | RMSE |
| W-MAE (5) | 104.2 | 113.6 | 121.1 | 123.6 | 101.9 | 39.8 | 0.551 |
| W-MSE (5) | 111.5 | 124.2 | 136.6 | 140.2 | 103.6 | 29.4 | 0.610 |
| SERA (5) | 108.0 | 143.4 | 249.1 | 348.5 | 166.4 | 34.8 | 0.872 |
| MAE (4) | 97.1 | 92.1 | 84.0 | 77.3 | 57.7 | 30.7 | 0.481 |
| MSE (5) | 96.7 | 91.6 | 84.2 | 78.5 | 59.5 | 29.2 | 0.487 |
| Ensemble | 109.6 | 129.9 | 164.2 | 166.5 | 102.5 | 29.6 | 0.626 |

    Finally, analogously to Fig. 7, we provide in Fig. 10 an intensity-scale diagram of the SEDI scores obtained by the ensemble
model. From comparing the figures on the finest scale (28 km) it is clear that the ensemble is able to substantially improve the
SEDI scores on the 50th–90th percentile thresholds, now comparable with SEDI scores for the W-MAE and W-MSE on these
thresholds (see Table 3), while scores between the 95th–99.9th percentile thresholds remain on-par with the SERA (5) model.
It is also interesting to note that, while scores do degrade with increasing scale on all thresholds except the 50th percentile, as
in Fig. 7, they degrade substantially less than in Fig. 7.



## 4   Discussion

The results presented in this paper indicate that the multi-layered convolutional long short-term memory (ConvLSTM) network can be adapted well to the task of spatio-temporal forecasting of extreme wind events by training the network with imbalanced regression loss. From Table 3, it is clear that the utilisation of the W-MAE, W-MSE or SERA loss results in substantially improved forecasts of extreme events of intensity thresholds between the 75th and 99th percentiles, as measured with the symmetric extremal dependency index (SEDI), as compared with either the standard mean squared error (MSE) or mean absolute error (MAE).

While superior SEDI scores were obtained by the SERA-trained model on intensity thresholds between the 90th–99th percentiles, we have shown that this is in large part due to a severely increased frequency bias, as well as increased coarseness, for extreme events in this range. As shown in Table 4, this bias can be greatly mitigated by merging the SERA (5) model with the W-MAE (5) and W-MSE (5) models into a joined ensemble. The ensemble limits frequency bias, limiting the tendency of the SERA (5) model to overshoot and forecast false alarms; it, furthermore, improves SEDI scores between the 50th–90th percentiles while remaining on-par with the SERA (5) model between the 95th–99.9th percentiles; overall RMSE is reduced (Table 4) and forecasts are spatially sharper (Fig. 8 and 9). It must be noted, however, that some of the high intensity hits made by the SERA (5) model are inevitably lost in the averaging process of the ensemble forecasts.

Although the inversely weighted MAE and MSE loss showed themselves to be less capable than the SERA loss in shifting performance towards the extreme thresholds, we do note that other weighting methods may yield better results. Shi et al. (2017), for example, utilised a linear weighting method for precipitation nowcasting using a trajectory gated recurrent unit (TrajGRU) network and reported improved performance at higher rain-rate thresholds as compared with the standard MSE and MAE (although this conclusion is based on the measuring of performance using the critical success index (CSI) and the Heidke skill score (HSS), neither of which is recommended in the literature for the forecast verification of rare events due to score degeneracy (Stephenson et al., 2008)).

We proceed to make a note on the spatial verification that was conducted using the minimum coverage method. Figure 7 suggests that upscaling the forecasts of the SERA (5) model offers no improvement to the forecasting of extreme events beyond the 75th percentile threshold. For non-extreme events of the 50th percentile threshold, forecasts improve at coarser scales as it becomes easier for the model to make correct predictions, which is the expected behaviour. This behaviour ceases, however, for events beyond the 75th percentile threshold. We suspect that this is due to the fact that when a spatio-temporal region of extreme events is missed by the model, it is typically missed completely and hence upscaling the forecast will do nothing to improve it. It may, furthermore, be explained by the fact that when forecasts of extreme events are made by the SERA (5) model, they are typically substantially coarser than the observation, and often forecasted when there are, in fact, no extremes present in the observation. We can see this well in the forecast visualisations provided in Fig. 8 and Fig. 9, or, additionally, in the supplementary material. Since we are using the minimum coverage criterion to spatially upscale the forecasts, a group of false alarms can easily result in an entire upscaled region being labelled as a false alarm, increasing the false alarm rate $F$ and thus degrading the SEDI skill score (see Eq. 7).



On another note, we wish to provide some insight into the predictions made by the multi-layered ConvLSTM network by discussing feature importance. In order to determine the importance of each of the 12 input frames that are used by the ConvLSTM to make its prediction, we proceeded to carry out a permutation test on the input data. For each input frame at time $T$ (-11–0), we randomly shuffle all input frames from the testset at time $T$ (essentially nullifying the information flow from this input frame), gather the model predictions from these permuted inputs and compute a skill score $S$ (in %) between the RMSE of the original prediction and target ($\mathrm{RMSE_{org}}$) and the permuted prediction and target ($\mathrm{RMSE_{perm}}$) i.e. $S = (1 - \mathrm{RMSE_{org}}/\mathrm{RMSE_{perm}}) * 100$. A score of 0 % indicates no change in RMSE, a score of 100 % indicates maximum increase in RMSE due to the permuted inputs and negative scores indicate decrease in RMSE due to the permuted inputs. Not only does this offer insight into the importance that each input frame carries in the ultimate prediction but it also helps to ensure that the model is, in fact, basing its predictions on the information flow between consecutive input frames rather than simply resorting to forecasting climatology. Figure A1 presents the RMSE skill scores obtained by the ensemble of the W-MAE, W-MSE and SERA-trained models as aggregated over the testset. The figure shows that root mean squared errors tend to get increasingly larger with permuted input frame $T$ approaching 0 hours, with an intermediate jump between -9 and -6 hours and a large jump from -3 to 0 hours, where a score of approx. 60 % is reached. This shows clearly that the predictions of the ensemble model are heavily dependant on the input frames, with the the most recent frames carrying most importance to the final prediction, but also, interestingly, the frames -9 to -6 hours, perhaps utilised by the ConvLSTM for the more long term dynamics. Most important to the final prediction is clearly the last input frame at time $T = 0$ hours, which comes as no surprise since the model is a regression model tasked to predict the continuation of spatio-temporal sequence from time frame $T = 0$ onward.

We finish by briefly discussing a number of possible extensions of this work. One disadvantage of utilising the entirety of available data in the context of this work is that many of the input-target samples containing extreme winds are samples where extreme winds are present in both the input as well as the target. Examples where there are no extremes present in the input, but the target is showing onsets of extremes, are disproportionately rare in the data although they clearly represent a more interesting problem (e.g. for early-warning systems). Improving our model as an early-warning system of onsets of extreme winds may thus be obtained by focusing model-learning on precisely such training samples, rather than employing all available samples. To this end, it could be worthwhile to change the model into a nowcasting model. This would entail reducing the lead-time of the model to below 6 hours while increasing the temporal and spatial resolutions of the data, possibly by utilising more precise ground data than raster data from satellites, as recommended by Amato et al. (2020).

This work may, furthermore, be extended by taking a multi-variate approach to wind speed forecasting whereby other atmospheric variables are included into the input of the model, which is an approach that is already being pursued in the community (see: e.g. Racah et al., 2017; Marndi et al., 2020; Xie et al., 2021). Marndi et al. (2020) suggest the utilisation of temperature, humidity and pressure into the forecasting task as these have been found to be "significantly more important than other [atmospheric variables]" - a result based on the work done by Cadenas et al. (2016). Xie et al. (2021) use these same three variables, as well as the 1-hour minimum and maximum temperature, while Racah et al. (2017) use a much larger set of 16 atmospheric variables, albeit for the classification of large-scale extreme weather events and not for regression of wind speed.





It may also be worthwhile to consider other atmospheric variables such as the convective available potential energy (CAPE) and deep-layer wind shear (DLS) due to their strong correlation with severe convective storm activity such as the occurrence of thunderstorms and supercells (see: e.g. Rädler et al., 2015; Tsonevsky et al., 2018; Chavas and Dawson II, 2021).

Finally, we note that while the ConvLSTM has proven itself to be highly effective at modelling complex spatio-temporal patterns, other models have since been proposed as promising improvements to the ConvLSTM for the task of spatio-temporal sequence forecasting. Most notably, the PredRNN and its predecessor PredRNN++, proposed by Wang et al. (2017) and Wang et al. (2018), respectively, have been demonstrated to be superior to the ConvLSTM for the task of video frame prediction by maintaining a global memory state rather than constraining memory states to each ConvLSTM module individually. Other alternative approaches include the usage of functional neural networks (FNNs) (see: Rao et al., 2020) or generative adversarial

networks (GANs) (see: Gao et al., 2020). Such models may well be of interest to the meteorological community pursuing data-driven, spatio-temporal forecasting.

## 5 Conclusions

In this paper we explored a deep learning approach to the task of spatio-temporal prediction of wind speed extremes in the short-to-medium range. To this end, we investigated the application of a multi-layered convolutional long short-term memory

(ConvLSTM) network, which we adapted to imbalanced spatio-temporal regression by training the model with either inversely weighted mean absolute error (W-MAE), inversely weighted mean squared error (W-MSE) or squared error-relevance area (SERA) loss. The models were trained and tested on reanalysis wind speed data from the European Centre for Medium-Range Weather Forecasts (ECMWF) at 1000 hP, providing multi-frame forecasts of horizontal near-surface wind speed over Europe with a 12 hour lead-time and in one hour intervals, using the preceding 12 hours as input. By standardising the data based on

the local wind speed distributions at each coordinate we focused the definition of an extreme event on its relative rarity rather than its absolute severity and considered extreme winds in terms of their local distributional percentile.

The model forecasts were verified by computing the symmetric extremal dependence index (SEDI) over various lead-times, spatial scales and intensity thresholds. After determining the optimal number of network layers for each of the models (trained with either W-MAE, W-MSE, SERA or standard MAE and MSE loss), a comparison between the different loss functions was

made in Table 3. The results show that the imbalanced regression loss functions investigated in this paper (W-MAE, W-MSE and SERA loss) can be used effectively to improve forecasting performance for extreme events beyond the 75th percentile threshold. While the results indicate superior performance of the SERA loss over the W-MAE and W-MSE loss in forecasting extreme wind events of intensity thresholds between to the 90–99th percentiles, we observed that this goes hand-in-hand with a severe frequency bias and an increased coarseness of the forecasts. While the SERA loss thus tends to produce worst-case

scenarios, we observe greatly improved results when combining the W-MAE, W-MSE and SERA-trained models into an ensemble. Table 4 and Fig. 10 show this quantitatively, while the forecast visualisations in Fig. 8 and 9 show qualitatively that the ensemble is able to model the complex spatio-temporal dynamics of both extreme and non-extreme wind speeds very effectively as far as 12 hours into the future. We conclude that the inversely weighted loss and the squared error-relevance area



loss provide relatively easy and effective ways to adapt deep learning to the task of imbalanced spatio-temporal regression and

its application to the forecasting of extreme wind events in the short-to-medium range, and may be best utilised as an ensemble.

With this work we hope to provide a valuable contribution to the area of deep learning for wind energy applications as well as

the area of imbalanced spatio-temporal regression and its verification as a forecasting problem.

*Code and data availability.* The current version of model is available from the project repository on Github: https://github.com/dscheep

ens/Deep-RNN-for-extreme-wind-speed-prediction under the MIT license. The exact version of the model used to produce the results

used in this paper is archived on Zenodo (DOI: 10.5281/zenodo.6796745), as are scripts to run the model and produce the plots for all the

simulations presented in this paper. The data used in this paper can be downloaded from the Copernicus Climate Change Service Climate

Data Store (CDS) of the ECMWF (see Hersbach et al., 2018), where the reanalysis data of the U and V components of the horizontal wind

velocity were taken at 1000 hPa from the *ERA5 hourly data on pressure levels from 1979 to present* dataset between years 1979-2021 (42

years) and between 40-56° N and 3-19° E. Scalar wind speed was obtained by computing the square root of the sum of the squares of the

two wind velocity components. Scripts to generate the data as such are available in our project code.

*Sample availability.* Sample forecasts are available at https://github.com/dscheepens/Deep-RNN-for-extreme-wind-speed-prediction/exa

mple_forecasts.

*Video supplement.* Video supplements are available at https://github.com/dscheepens/Deep-RNN-for-extreme-wind-speed-prediction/exa

mple_forecasts.



**Appendix A: Figures**

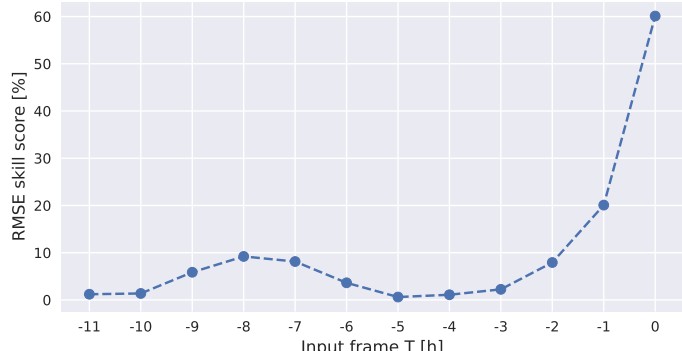

**Figure A1.** Results from the permutation test as carried out for the ensemble model consisting of the W-MAE, W-MSE and SERA-trained ConvLSTM networks. The figure shows the RMSE skill score (in %) between the targets and the normal predictions of the ensemble, and the targets and the predictions resulting from randomly permuting the inputs at time-frame $T$. A score of 0 % indicates no change in RMSE, a score of 100 % indicates maximum increase in RMSE due to the permuted inputs and negative scores indicate decrease in RMSE due to the permuted inputs.





*Author contributions.* DS and KHS conceptualised the research. DS carried out the data curation, formal analysis, investigation, methodology, programming, validation, visualisation and writing of the paper. KHS and IS provided supervision, scientific discussion and guidance, and reviewed and revised the work. IS and CP carried out project administration and CP also provided computing resources.

*Competing interests.* The authors declare that they have no conflict of interest.

*Acknowledgements.* Our special thanks go to Markus Dabernig and Aitor Atencia (Zentralanstalt für Meteorologie und Geodynamik, Vienna) who provided invaluable and extensive feedback to the paper at its final stages. This work was funded within the Austrian Climate and Research Programme (ACRP) under the project name MEDEA (KR19AC0K17614) to further research on renewable energy and meteorologically induced extreme events.





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
