# Peer review of "Adapting a deep convolutional RNN model with imbalanced regression loss for improved spatio-temporal forecasting of extreme wind speed events in the short-to-medium range"

_EGUsphere, 2022_

## Author Response (AR1)

**Author's Response**

**Point-to-point response to Referee #1:**

Dear Referee,

Thank you for your review and suggestions for improving the clarity of the manuscript. We proceed here to answer your questions to the best of our ability.

Major comments:

1.
    1. &
    2. We have addressed the points about model application and motivation better in the introduction. The aim of the paper is to investigate how the spatio-temporal predictions of a deep learning forecasting model may be improved for the extremes through the manipulation of loss function. Indeed, it is likely that any improvements for the extremes go hand-in-hand with predictions deteriorating for non-extremes. We attempt to show whatever trade-offs there are in this regard more clearly in the results. From an application perspective, such a trade-off would nevertheless be very attractive to improve a model's efficacy as a warning-system: Its ability to distinguish different levels of extremes would be critical while its ability to distinguish different levels of non-extremes would be irrelevant.

        The focus on relative rarity in our definition of an extreme event is now better motivated in the methodology. What we mentioned perhaps too briefly in our manuscript is that the primary reason for making this choice stems from the fact that extreme winds in the absolute sense (e.g. exceeding 25 m/s) occur exclusively off-shore in a very localised region and are thus absent from the majority of coordinates in the ERA5 reanalysis data (where typically only max. wind speeds of 8-10 m/s are present). Defining extreme events rather in terms of their relative rarity at each coordinate allows us to investigate forecasting improvements of extreme events more generally by looking at improvements on the tails of the respective distributions, regardless of what absolute values these tails actually obtain. Demonstrating that forecasting performance on the tails can be improved in this more general context, by adapting the loss function, can be swiftly translated to other cases where the tails of the distributions denote actual hazardous events. This paper serves to help indicate as to how the loss function may be adapted as to best help improve forecasting performance on the tails. indeed, with the assumption that the tails typically denote extreme events of some form. The paragraph in question in the methodology has been modified accordingly.

    3. It is true that probabilistic output is typically preferred by users but here we would argue that in the same way as how deterministic NWP forecasts are commonly aggregated into probabilities by utilising large ensembles, the deterministic forecasts of the ConvLSTM regression model can be aggregated into probabilities e.g. with different ensemble members trained on different subsets of the training set.

2.  We certainly understand the reviewer's concern. However, instead of providing benchmark comparisons with other models, this point has made it apparent that we have had to more clearly state the aim of the paper in the introduction. The aim of the paper is not to investigate our adaptation of the ConvLSTM as compared with other state-of-the-art models. Rather, the aim of the paper is to investigate how the performance of a popular spatio-temporal deep learning model (the ConvLSTM) changes for various thresholds of extreme events by utilising two types of loss functions proposed in the literature on imbalanced regression. For literature on the improvements of the ConvLSTM over other state-of-the-art models, or simpler non-convolutional, or non-recurrent models, the reader could be referred to Shi et al. (2015) or Shi et al. (2017), which is now mentioned in the introduction.

3.  This point is now clarified in the methodology. 1000 hPa wind speed is the only variable used. The model takes in 12 consecutive hours of wind speed data over the 64x64 grid, comprising a tensor of size 12x64x64. This tensor is encoded through the encoding network into a hidden state and decoded through the decoding network into an output tensor of size 12x64x64 comprising the forecast of the subsequent 12 hours. The temporal correlations between consecutive hours are taken into account implicitly by the convolutional LSTM layers. For the exact details on how the convolutional LSTM modules achieve this the paper refers to Shi et al. (2015).

Minor comments:

1.  Done.

2.  Thank you for the comment. Indeed, this is missing in the literature review. We have updated the introduction with a paragraph on probabilistic forecasts.

3.  The review of data-driven weather forecasting models has been removed from the introduction as we have come to see that this is, indeed, not the focus of the manuscript.

4.  For this study initially 5 different pressure levels, including the diagnostic 10 m wind fields, were investigated. With the currently implemented hub heights of wind turbines in Austria of 100 to 135 m above ground level, the 1000 hPa fields are more appropriate (corresponding to ca. 100-130 m in Eastern Austria (main wind energy region)). Furthermore, reanalysis methods typically interpolate across- and output data at different pressure levels rather than height levels, which also motivates the choice. We have modified the methodology accordingly.

5.  Thank you for pointing this out. We have decided to repeat the experiments utilising a Yeo-Johnson power transform (Yeo and Johnson, 2000) before the zero-mean, unit-variance normalisation in order to make the distributions more Gaussian-like.

6.  Noted. We have included another, linear, weighting method in the comparison and include the SERA loss with three different sets of control-points in order to provide a more complete comparison.

7.  Thank you - this was indeed overlooked. After much deliberation we have decided to replace the SEDI score with a simpler set of scores that are substantially more wide-spread in the literature as well as easier to interpret: H, FAR, TS and B. Furthermore, the RMSE has been included in the analysis to provide an indication of the typical magnitudes of errors, which the categorical scores are unable to provide.

8.  In principle it would be possible to incorporate any particular loss function into the model training. This was out of scope for this work, however. Machine learning based methods, as well as also statistical methods, tend to smoothen the forecasts and underestimate especially the tails. The idea was to implement a loss function which is able to account for that, sharpen the forecasts and is able to get the intensities in the right order. This is essential for not only wind energy applications

(planning of feed-in, curtailment, etc.) but also for e.g. tourism (winter sports), transportation, and forestry. We mention this possibility in the discussion.

9. Rather than averaged, the results are aggregated over all lead-times. This has been clarified in the new manuscript.

10. The ensemble has been removed from the comparison. We have decided that it does not, in fact, provide any useful information towards the comparison of the loss functions.

11. Full fields. This has been clarified in the new manuscript.

12. Because Fig. A1 is a comparison of the continuous-valued wind speed fields and as such requires a continuous score like the RMSE to compare. The SEDI is a categorical score that can only be used with discrete data considering events and non-events i.e. after applying a threshold to the continuous wind speed fields.

**Point-to-point response to Referee #2:**

Dear Referee,

Thank you for your review and suggestions for improving the clarity of the manuscript. We proceed here to answer your questions to the best of our ability.

Major comments:

1. There seems to be a slight misunderstanding here. The integral in eq. 5 goes over thresholds t in [0,1] where t=0 takes into account all datapoints. Only then, e.g. at t = a only those points with relevance >= a are included in the integral. At increasingly higher thresholds, increasingly many points are, indeed, discarded from the computation but they are not absent from the final integral. We will make sure, however, to clarify this subtlety in the methodology. We will also make sure to clarify the differences between the SERA and the re-weighing of the MAE or MSE, but do wish to highlight their common goal, which is to increase the importance of the tails in the loss function. For this reason we do not see that a comparison of the two is in any way unsound or unfair. Indeed, we would argue that the fact that these two methods attempt to achieve the same goal by different means is surely what makes the comparison of interest in the first place.

2. We are not sure whether we fully understand this point as the model that we investigate outputs deterministic predictions, not probabilities. We do, however, acknowledge that model errors tend to increase for the distributional tails due to larger absolute values. Having said that, for a prediction to be correct categorical scores of binary events require only that a prediction and observation pair both surpass some threshold t regardless of how largely t was in fact overshot. In terms of discrete extreme event prediction, the continuous errors are thus irrelevant. We have decided, however, to include in the results a continuous score (the RMSE) between the continuous prediction and observation fields, and present its variation between the same set of thresholds used for the categorical scores, to show how the continuous errors differ between the different models i.e. loss functions.

3. Noted. We have decided to include another, linear, weighting method in addition to the inverse weighting, and will provide results of the SERA loss with its lower control-point set to either the 90th, 75th or 50th percentile while keeping its upper control-point fixed at the 99th percentile. We have also gone to greater lengths to compare and contrast the results of the different loss functions in order to provide a more complete picture.

4. Noted. The literature review on DL methods has been removed from the introduction, as, indeed, this is not the main focus of the manuscript. Instead, additions have been made to the review of extreme event predictions and another paragraph has been added in which the aim and the motivation of the paper are more clearly stated.

5. The ensemble was constructed by averaging the individual predictions, equally weighted. However, we have decided to remove the ensemble from the comparison all together as it does not, in fact, provide any useful information towards the comparison of the loss functions.

6. Noted. We propose changing the title as follows: "Adapting a deep convolutional RNN model with imbalanced regression loss for improved spatio-temporal forecasting of extreme wind speed events in the short-to-medium range.".

Minor comments have been incorporated in the new manuscript. One other question was asked, however:

Q: Table 2: what happens if the architectures become even more complex?
A: We will first mention that there is an architectural limitation to the number of possible layers in the network due to the input dimensions halving and the hidden dimensions doubling with each additional layer (due to the adopted auto-encoder network structure): This puts the upper limit at 7 layers for inputs fields of shape 64x64. We decided to stop at 5 layers, however, due to limitations in computational capacity (as shown in Table 1, the number of parameters grow rapidly with increasing network layers). For reference, running the entire experiment with 2—5 layers on the GPU of the university cluster took ca. 2 weeks of computing time. Finally, it is evident from Table 2 that the validation loss does not change substantially anymore from 4 to 5 layers and thus we would not expect to see any substantial changes for 6 or 7 layers either.

**List of all relevant changes:**

With this response we wish to clarify all relevant changes made to the manuscript.

1. The Introduction (Section 1) has been reworked considerably to reflect better the aim of the paper, which is to investigate how the spatio-temporal predictions of a deep learning forecasting model may be improved for extreme events through the manipulation of the loss function. To this end, an additional paragraph has been appended at the end of the introduction to better clarify and motivate the aim of the paper. This also mentions applications.
2. The rather extensive paragraph in the introduction that discussed different DL models and architectures has been scrapped completely as this, indeed, is not the focus of the paper (although some references have been moved to the discussion as potential improvements to the ConvLSTM). Instead, more consideration has been given to the problem of imbalanced regression, particularly in the context of deep learning. This includes some of the references suggested by the referees. Also included is a paragraph on probabilistic forecasting methods (as proposed by Referee #1).

3. The Methodology (Section 2) has also been reworked considerably. In Section 2.1, the focus on relative rarity rather than absolute severity has been better motivated, as well as the usage of 1000 hPa wind fields. In Section 2.1 we also explain the new standardisation strategy with which we have repeated our experiments: The data have now been preprocessed with a Yea-Johnson power transform to make the local wind speed distributions more Gaussian-like, before standardising using zero-mean, unit-variance normalisation (as pointed out by Referee #1).
4. Section 2.2 has been changed to provide a better explanation of how the 12-hour inputs are processed through the ConvLSTM network to create 12-hour forecasts. The subsection previously named 'implementation' (Section 2.2.3) has now been embedded into Section 2.2. Subsection 2.2.1 has been renamed 'Weighted Loss' and now describes the inverse weighting method as well as a linear weighting method. The linear weighting is included in the analysis to obtain a better understanding of how such different weighting results in different forecasting performance. Subsection 2.2.2 has only been changed insofar as to explain now that the analysis of this paper includes the SERA loss implemented with three different primary control-points (with the secondary control-point fixed), in order to get a better idea of how the choice of control-points affects forecasting performance.
5. Section 2.3 has been modified considerably. It was decided that the spatial verification provided only minimal additional information to the analysis of the results and as such needlessly complicated the methodology. Analysis over spatial scale using the minimum coverage method has thus been omitted. Furthermore, due to the complications alluded to by Referee #1, the symmetric extremal dependence index (SEDI) score has been replaced by a simpler set of scores that are substantially more wide-spread in the literature as well as easier to interpret: H, FAR, TS and B. Furthermore, the RMSE has been included in the analysis to provide an indication of the typical magnitudes of errors, which the categorical scores are unable to provide. Lastly, the section title 'Forecast Verification' has been changed to 'Forecast Evaluation' as the word 'verification' is, indeed, rarely used outside of atmospheric science for the same purpose (as pointed out by Referee #2).

6. With all models having been retrained on the updated data (preprocessed with a Yeo-Johnson power transform), the Results (Section 3) has been updated accordingly. As

suggested by Referee #2, the permutation test results have been moved from the Appendices to the Results section. The original single figure has been split into four plots in order to easily compare the investigated loss functions within four groups: Inverse weighting, linear weighting, SERA loss and the standard MAE or MSE loss. Additionally, an extra figure has been added to the results (Fig. 7), which provides a comparison of the forecast distributions obtained from each loss function.

7. With the aim of the paper focussed specifically on the comparison of the different loss functions investigated here, it was decided that the previously presented ensemble model no longer provided any information of interest to that aim, and thus was removed.

8. The forecast examples have been chosen anew and are now presented in terms of the different local percentile thresholds, rather than the actual wind speed. This was decided to be a better representation since this is how the forecasts were evaluated throughout the paper. With this representation it is straightforward to see precisely where which type of event was forecasted by each model.

9. As suggested by Referee #2, the title has been changed to be more descriptive. We propose: "Adapting a deep convolutional RNN model with imbalanced regression loss for improved spatio-temporal forecasting of extreme wind speed events in the short-to-medium range."

We hope that with these changes, the paper now provides a clearly motivated and rigorously conducted investigation into three common manipulations of the loss function for the forecasting of extreme events using deep learning, with which we hope provides a valuable contribution to the to the area of deep learning for spatio-temporal imbalanced regression and its application to wind energy forecasting research.